# Pharmacokinetic Analysis of an Isoniazid Suspension Among Spanish Children Under 6 Years of Age

**DOI:** 10.3390/antibiotics14010074

**Published:** 2025-01-12

**Authors:** Antoni Noguera-Julian, Emma Wilhelmi, Maria Cussó, Rob Aarnoutse, Angela Colbers, Loreto Martorell, Maria Goretti López-Ramos, Joan Vinent, Rosa Farré, Dolors Soy, Sílvia Simó-Nebot, Clàudia Fortuny

**Affiliations:** 1Malalties Infeccioses i Resposta Inflamatòria Sistèmica en Pediatria, Servei d’Infectologia, Institut de Recerca Pediàtrica Sant Joan de Déu, 08950 Barcelona, Spain; maria.cusso@sjd.es (M.C.); silvia.simo@sjd.es (S.S.-N.); claudia.fortuny@sjd.es (C.F.); 2Departament de Cirurgia i Especialitats Medicoquirúrgiques, Facultat de Medicina i Ciències de la Salut, Universitat de Barcelona, 08036 Barcelona, Spain; 3Centro de Investigación Biomédica en Red de Epidemiología y Salud Pública (CIBERESP), 28029 Madrid, Spain; 4Pharmacy Department, Hospital Sant Joan de Déu, 08950 Barcelona, Spain; emma.wilhelmir@sjd.es (E.W.); mariagoretti.lopez@sjd.es (M.G.L.-R.); joan.vinent@sjd.es (J.V.); rosa.farre@sjd.es (R.F.); 5Department of Pharmacy, Radboud University Medical Center, Radboud Institute for Health Sciences, 6525 GA Nijmegen, The Netherlands; rob.aarnoutse@radboudumc.nl (R.A.); angela.colbers@radboudumc.nl (A.C.); 6Molecular Genetics Department, Hospital Sant Joan de Déu, 08950 Barcelona, Spain; loreto.martorell@sjd.es; 7Pharmacy Department, Division of Medicines, Hospital Clinic of Barcelona, University of Barcelona, 08036 Barcelona, Spain; dsoy@clinic.cat

**Keywords:** acetylator status, child, isoniazid, pharmacokinetics, suspension, tuberculosis

## Abstract

**Background**: Isoniazid (INH) remains a first-line drug for the treatment of tuberculosis (TB) in young children. In 2010, the WHO recommended an increase in the daily dose of INH up to 10 (7–15) mg/kg. Currently, there are no INH suspensions available in Europe. **Methods**: We aimed to characterize the pharmacokinetics of a licensed INH suspension (10 mg/mL, Pharmascience Inc., Montreal, QC, Canada) in children receiving INH daily at 10 mg/kg in a single-center, open-label, non-randomized, phase IIa clinical trial (EudraCT Number: 2016-002000-31) in Barcelona (Spain). Samples were analyzed using a validated UPLC-UV assay. The *N*-acetyltransferase 2 gene was examined to determine the acetylation status. A non-compartmental pharmacokinetic analysis was conducted. **Results**: Twenty-four patients (12 females) were included (primary chemoprophylaxis, *n* = 12; TB treatment, *n* = 9; and TB infection preventive treatment, *n* = 3). The acetylator statuses were homozygous fast (*n* = 3), heterozygous intermediate (*n* = 18), and homozygous slow (*n* = 2; unavailable in one patient). The INH median (IQR) C_max_ and AUC_0–24h_ values were 6.1 (4.5–8.2) mg/L and 23.0 (11.2–35.4) h∙mg/L; adult targets (>3 mg/L and 11.6–26.3 h∙mg/L) were not achieved in three and six cases, respectively. Gender, age at assessment (<2 or >2 years), and INH monotherapy (vs. combined TB treatment) had no impact on pharmacokinetic parameters. Significant differences in C_max_ (*p* = 0.030) and AUC_0–24h_ (*p* = 0.011) values were observed based on acetylator status. Treatment was well tolerated, and no severe adverse events were observed; three patients developed asymptomatic mildly elevated alanine aminotransferase levels. **Conclusions**: In infants and children receiving a daily INH suspension at 10 mg/kg, no safety concerns were raised, and the target adult levels were reached in the majority of patients.

## 1. Introduction

Tuberculosis (TB) remains a health challenge worldwide, especially in low- and middle-income countries. Despite being a preventable disease, TB is estimated to have affected 1.25 million children and adolescents under 15 years of age globally in 2022, with 47% of cases occurring in those under 5 years old [1]. In Spain, a low-burden country, the pediatric TB incidence rate in 2022 was three cases per 100,000 inhabitants, with 44% of cases also in children under 5 years of age [2]. TB caused 214,000 deaths in the pediatric age group, with three-quarters occurring in the youngest children, where the treatment coverage gap remains the largest [1]. In the latter, the risk of progression from primary TB infection to disease inversely correlates with age, TB usually develops within the first year after primary infection, and severe and disseminated forms are more frequent than in older children or adults [3,4].

Significant progress has been made over the past decade, when the first edition of the *Roadmap Towards Ending TB in Children and Adolescents* was published [5], including advocacy for TB policy inclusion, the recognition of pediatric-specific needs, the formation of national TB working groups, improvements in diagnostic and treatment strategies, and the availability of pediatric TB formulations. In contrast, several remaining challenges requiring urgent action have been identified, including the need for pharmacokinetic and safety data to enable children to benefit from shorter regimens for both TB infection preventive treatment and TB disease, including drug-resistant TB [1].

Isoniazid (INH) exerts a bactericidal effect by inhibiting mycolic acid synthesis, thereby disrupting cell wall formation. It remains a first-line treatment for all forms of drug-susceptible TB, including in newer, shorter regimens, and is used in both the intensive and continuation phases [6]. High-dose INH (e.g., 20 mg/kg daily) may also be used in select cases of drug-resistant TB. INH is included in all preferred regimens for TB infection preventive treatment, including shorter rifapentine-containing regimens [7,8]. INH also remains the most widely used drug for primary chemoprophylaxis in young children or immunosuppressed patients following contact with a smear-positive TB index case [7].

Appropriate dosing and adherence to treatment are essential to mitigate acquired antibiotic resistance, as cases of drug-resistant TB continue to rise over time [9,10]. Given the limited pharmacokinetic data linking drug concentrations to clinical outcomes in pediatric TB, target exposures from adult studies are used as surrogate markers for optimal anti-TB therapy dosages in children. The World Health Organization (WHO) revised the recommended dosages for first-line oral anti-TB medications in children in 2009. This update was based on a systematic review of studies showing that children need higher doses than adults to achieve equivalent serum drug concentrations [11]. The daily dose of INH was raised from 5 mg/kg (range: 4–6 mg/kg) to 10 mg/kg (range: 10–15 mg/kg), with a daily maximum of 300 mg [11]. Later, the INH dosing range was further extended to 7–15 mg/kg [12].

Maturation factors in early childhood significantly affect the pharmacokinetic parameters of anti-TB drugs in children [13]. Specifically, INH levels are influenced by genetic polymorphisms of N-acetyltransferase type 2 (*NAT*2) [14]. Two recent systematic reviews and meta-analyses have evaluated the effectiveness and pharmacokinetic profiles of first-line oral drugs for treating drug-susceptible TB in children and adolescents, following the 2009 WHO-recommended dosages [15,16]. Nearly all studies included in these analyses were conducted in high TB burden regions and reported significant rates of malnutrition, HIV co-infection, and other comorbidities.

This single-center, non-randomized, open-label, phase IIa clinical trial aimed to determine the pharmacokinetics of a licensed INH suspension and assess the effects of age and *NAT*2 acetylator status in children under 6 years in Spain, a low TB burden country.

## 2. Results

From January 2018 to February 2020, 41 children < 6 years of age were invited, and 27 of them accepted to participate in the trial. Informed consent was withdrawn for three patients enrolled in Group A before the day of the pharmacokinetic assessment, two of whom received at least one dose of the study medication. A total of 24 patients (12 females, 50%; 7 in Group A) completed the study protocol and were included in the pharmacokinetic analysis (Table 1).

Isoniazid was indicated as part of TB disease and TB infection preventive treatment in 9 and 3 patients, respectively, and as primary chemoprophylaxis after contact with a smear-positive TB index case in the remaining 12 children. A total of 12 patients (4 in Group A) were treated with INH monotherapy, and 12 received INH in combination with rifampicin (3 in Group A). None of the patients were HIV-infected. No comorbidities were observed except for one 5-year-old girl affected with neuroblastoma who was receiving INH monotherapy for the preventive treatment of TB infection, together with immunotherapy. Baseline ALT, hemoglobin, and albumin levels were within normal limits in all cases.

On the day of the pharmacokinetic assessment, the median (IQR) age was 2.8 (1.8–4.2) years, including 7 and 17 patients in Groups A and B, respectively. All participants had been receiving INH in licensed Spanish formulations for a median (IQR) duration of 12.4 (10.1–25.3) weeks at the time of study drug initiation. Subsequently, the duration of treatment with the study drug prior to the pharmacokinetic assessment was 6 (4–6) days for patients on INH monotherapy and 8 (7–10) days for those receiving additional anti-TB drugs. The adherence to the study drug in the previous 3 days was 92%, and the tolerance to treatment was optimal in all cases. According to the *NAT*2 genotype, the acetylator status was FF, FS, and SS in 3, 18, and 2 patients, respectively; the acetylator status could not be determined in one child.

The estimated pharmacokinetic parameters are outlined in Table 2, revealing substantial inter-individual variability. INH C_max_ values ranged from 1.8 to 10.2 mg/L, with the adult-recommended target of >3 mg/L achieved in 20 out of 23 cases (87.0%). The median (IQR) AUC_0–24h_ of INH was 23.0 (11.2–35.4) h∙mg/L. The recommended AUC_0–24h_ range in adults (11.6–26.3 h∙mg/L) was not achieved in 6/23 patients (26.1%). This was significantly associated with the FF *NAT*2 acetylator genotype, occurring in all 3 FF patients versus 2 of 19 FS/SS patients, both FS (*p* = 0.004).

Gender, age at assessment (<2 or ≥2 years), and concomitant use of other anti-TB drugs had no impact on pharmacokinetic parameters. Significant differences in C_max_, AUC_0–24h_, *t*_1/2_, Cl/F, and Vd/F values were observed based on acetylator status (Table 3, Figure 1A–C). A trend towards higher C_max_ and AUC_0–24h_ values with longer pre-dose fasting times was observed (*r* = 0.283, *p* = 0.053 and *r* = 0.257, *p* = 0.086, respectively). The post-dose fasting time had no impact on the pharmacokinetic parameters.

Treatment was well tolerated in all cases. No severe adverse events were observed. Eleven mild adverse events unrelated to the treatment were recorded in 8 out of 26 (30.8%) patients (Table 4). Three patients developed asymptomatic mild hypertransaminasemia (range: 61–204 IU/L) that normalized after treatment interruption. Neither adverse events nor hypertransaminasemia were associated with the *NAT*2 genotype.

## 3. Discussion

In this open-label clinical trial, we analyzed the pharmacokinetics of revised WHO-recommended doses of INH using a licensed oral solution in young Spanish children. This study is one of few evaluating INH pharmacokinetics in children in a low TB burden, high-income country, with unique covariate differences from high TB burden regions that are known to influence anti-TB drugs pharmacokinetics, such as an ethnically diverse cohort, the absence of malnourished or HIV co-infected patients, and primary chemoprophylaxis post-exposure to a smear-positive TB index case being the most common indication of INH [15,16,19]. This study reflects the nature of the pediatric TB epidemic in low-burden countries like Spain, where active contact tracing is routine, most infected young children are detected in early infection stages, and outcomes are generally excellent [20].

When this study was planned and conducted, Europe lacked child-friendly formulations of first-line oral anti-TB drugs, aside from rifampicin suspension [21]. In Spain, pediatricians had to rely on off-label alternatives, resulting in uncertain final dosages, until the pediatric dispersible fixed-dose combination tablets of first-line anti-TB drugs were finally made available in July 2023 [22]. The latter include 100 mg INH tablets, 100 mg ethambutol tablets, 50 mg INH + 75 mg rifampicin tablets, and 50 mg INH + 75 mg rifampicin + 150 mg pyrazinamide tablets and are now routinely used for all indications in young Spanish children. We used a Canadian-licensed INH suspension (PDP-Isoniazid suspension, 10 mg/mL; Pharmascience Inc., Montreal, QC, Canada) for this study. Although the formulations of anti-TB drugs received by the patients are not specified in the aforementioned meta-analyses, first-line anti-TB drug suspensions are infrequently used to treat pediatric TB in high-burden, low-income countries due to large volumes for adequate dosing, cold chain requirements, shorter shelf-life and stability, and the absence of liquid fixed-dose combinations [15,16]. Nevertheless, suspensions are easy to administer and useful for children requiring precise dosing, especially the youngest infants, as well as for patients who find solid formulations challenging to ingest. The new dispersible fixed-dose combination tablets dissolve easily in water, enabling administration in liquid form to children. Qualitative research has shown that these tablets are palatable and compare favorably to other formulations for treating pediatric TB, although pragmatic challenges were also identified [23].

Since the WHO’s 2009 recommendation to double the daily INH dose from 5 to 10 mg/kg of body weight, several studies have been conducted to validate the new dosage across the entire pediatric age range, from premature babies to adolescents [15,16,24,25,26,27]. In line with other authors, we observed median (IQR) C_max_ (6.1 [4.5–8.2] µg/mL) and AUC_0–24h_ (23.0 [11.2–35.4] h∙mg/L) values comparable to the targets reported in the adult literature (3 to 6 µg/mL and 11.6–26.3 h∙mg/L) [17,18]. However, we noted substantial interindividual variability, with C_max_ and AUC_0–24h_ values falling below or above the recommended range in 13.0%/26.1% and 26.1%/39.1% of patients, respectively. The small sample size in our study likely limited our ability to observe the age-related differences in pharmacokinetic parameters reported by other authors, which have been attributed to higher drug clearance per kilogram in younger patients (below 2 years of age) [16,28,29]. None of the children in our study were malnourished or HIV-infected, both of which have also been associated with lower INH exposure [16,18,19]. The administration of INH on an empty stomach is recommended to enhance absorption [30]. We observed a near-significant relationship between pre-dose fasting time and both C_max_ and AUC_0–24h_ values, despite the latter being known to be less sensitive to the effect of food. Total exposure over time, expressed as AUC_0–24h_, is probably more relevant to the bactericidal activity of INH and the risk of *Mycobacterium tuberculosis* mutation emergence than C_max_, according to animal and hollow fiber models [31,32]. Therefore, INH administration in a fasting state remains desirable, though this may not be always feasible in some infants and toddlers.

The current WHO-recommended doses of rifampicin in children result in lower drug exposures than in adults, prompting advocacy for updated dosages, new fixed-dose combinations, and revised weight bands [25,33,34]. In contrast, the INH dose appears sufficient in most cases [15,25]. The elimination of INH is strongly influenced by the trimodal *NAT*2 acetylation pathway and its maturation from birth, including in premature infants [14,26]. As previously reported, six children in our study did not achieve the recommended AUC_0–24h_ range, which was associated with the *NAT*2 FF acetylator genotype. This finding is likely of limited relevance in our setting, where most children were treated preventively or for non-severe TB. However, insufficient drug exposure may lead to poorer outcomes in patients from high-burden countries where (a) treatment is calculated based on weight bands, which may result in lower doses per kg for some children in the upper weight range; (b) extrapulmonary and severe disease are common, and adequate INH exposure is critical for effective treatment; and (c) other risk factors for lower exposure, such as malnutrition and HIV co-infection, are prevalent.

Conversely, 39% of patients exhibited INH AUC_0–24h_ values above the recommended range, placing them at a higher risk of toxicity. Hepatitis is the most common adverse event attributed to INH and is idiosyncratic, not clearly related to the dose or duration of therapy [35]. In previously healthy children, INH-associated hepatitis is typically mild, non-symptomatic, and self-limiting, as observed in our study and reported previously with both the old and new WHO-recommended doses [36,37,38,39]. Apart from mild hepatitis, the study drug was well tolerated in our study, with only mild clinical adverse events observed, none of which were considered causally related to INH suspension. If available, *NAT*2 genotyping could be useful for individualizing INH doses in children at higher risk of treatment failure or increased toxicity [40]. In adults, a genotype-based dosing trial has already demonstrated improved clinical outcomes and safety [41].

Besides the small sample size and the non-randomized, open-label design, our study has additional limitations. We could not compare the bioavailability of the study drug with other INH formulations. The sample was unbalanced regarding *NAT*2 acetylator status, limiting conclusions about pharmacokinetic variability. Additionally, only three samples were collected per patient, which restricts the optimal estimation of pharmacokinetic parameters. This limited sampling scheme was chosen based on a balance between obtaining sufficient pharmacokinetic data and minimizing the burden on young participants. The impact of body weight on INH pharmacokinetics was not analyzed, as all study participants had weights within the normal range, with no cases of malnutrition. Drug administration can be challenging in children, and some medication loss from spillage or spitting cannot be entirely excluded. We were also unable to assess the relationship between drug exposure and treatment outcomes, nor the long-term tolerability and safety of the study drug. Finally, our results may not be generalizable to high-burden TB settings due to differences in patient characteristics.

## 4. Materials and Methods

### 4.1. Design and Setting

A single-center, non-randomized, open-label, phase IIa clinical trial (EudraCT number: 2016-002000-31) was conducted in Hospital Sant Joan de Déu, a pediatric referral hospital in Barcelona, Spain. The study protocol was approved by the local Ethics Committee (ref. AC 12-16) and conducted in accordance with the Declaration of Helsinki and Good Clinical Practice guidelines. At enrollment, written informed consent for participation was obtained from the parents or legal guardians of all participants.

### 4.2. Study Procedures

Children aged 1 month to 6 years, for whom daily oral INH (10 mg/kg/day) treatment was indicated or already initiated, were invited to participate in this study. Treatment with INH was indicated for primary chemoprophylaxis after contact with a smear-positive TB index case, for the preventive treatment of TB infection, or as part of TB disease treatment alongside other anti-TB drugs, in accordance with national guidelines [30]. In Spain, a diagnosis of TB infection is made for patients who have a positive immunodiagnostic TB test but lack clinical or radiological signs and symptoms indicative of TB. In contrast, a diagnosis of TB disease is established based on a combination of epidemiological, clinical, radiological, and microbiological criteria, as outlined in published consensus guidelines [42]. Children were not eligible if they had baseline alanine aminotransferase (ALT) levels > 50 IU/L (prior to INH administration), or if they were affected by any other infectious, renal, hepatic disease or any other condition that could alter INH metabolism, or if they were being treated with drugs known to potentially affect INH metabolism.

Data collected from all participants included gender, ethnicity, indication for INH treatment, and concurrent drug use. For breastfeeding patients, it was noted whether the mother was also receiving INH. Baseline measurements included ALT, hemoglobin, and albumin levels. On the day of pharmacokinetic sampling, age, weight, and nutritional status (weight-for-length/height Z-score in children < 5 years of age, and body mass index-for-age Z-score in children > 5 years of age, as per the WHO Child Growth Standards, available at: https://www.who.int/tools/child-growth-standards; accessed on 25 October 2024) were recorded.

### 4.3. Drug Administration, Pharmacokinetic Sampling and Laboratory Analysis

At inclusion, INH treatment was initiated or switched to PDP-Isoniazid suspension (10 mg/mL, Pharmascience Inc., Montreal, QC, Canada) at the standard daily dose (10 mg/kg/day) and with recommended fasting periods (at least 3 h pre-dose and at least 30 min post-dose), in accordance with national guidelines [30]. Adherence to anti-TB treatment is routinely assessed using the Eidus–Hamilton test and a written survey at each visit [10]. For this particular study, parents were asked to complete a specially designed written questionnaire on each of the three days preceding the pharmacokinetic assessment. In case of exclusive breastfeeding, HIV co-infection, or malnutrition, supplementation with pyridoxine (1–2 mg/kg/day) was recommended.

The pharmacokinetic study day was scheduled after a minimum of 3 days of treatment with the study drug for INH monotherapy or 5 days for combined treatment with other anti-TB drugs. At admission, a peripheral venous catheter was placed after the administration of local anesthetic. A 10 mg/kg dose of the study drug was precisely measured by the nurse using a graduated syringe to the nearest decimal point, with the dose rounded to the nearest tenth of a milliliter (e.g., a 5.63 kg infant would receive 5.7 mL of INH suspension), and administered orally under the investigators’ supervision.

The INH dose, the time of INH administration, and the fasting periods before and after INH administration were recorded. Three venous blood samples, each ranging from 0.5 to 1 mL, were collected through the venous catheter at 1, 3, and 6 h post-dose or at 2, 4, and 8 h post-dose; children were alternately assigned to one of the schedules based on the order of inclusion. After completion of the study treatment, the participants’ care was transitioned to the local TB program.

Serum separator tubes were used to collect blood samples, which were immediately placed on ice and centrifuged within 30 min. Subsequently, 200 µL aliquots of serum were frozen at −80 °C and shipped to the Netherlands on dry ice. INH concentrations were determined through a validated assay involving liquid-liquid extraction and ultra-performance liquid chromatography with UV detection. The method demonstrated an accuracy range of 97.8% to 106.7%, varying with concentration levels. The within-day and between-day variation coefficients were under 13.4% and 3.2%, respectively, across a concentration range of 0.05 to 15.1 mg/L. The assay’s lower limit of quantification for INH was 0.05 mg/L.

DNA was extracted from the remaining blood cells using standard protocols for *NAT*2 genotyping [43]. Samples were screened for known polymorphisms within the *NAT*2 gene. Primers were designed based on the *NAT*2 gene RefSeq accession number NM_000015.2. The complete 873 coding nucleotides were amplified by PCR and sequenced through capillary electrophoresis using an ABI Prism 3130 Genetic Analyzer (Applied Biosystems Inc., Foster City, CA, USA). Sequencing data were analyzed with Sequencing Analysis software version 6 (Applied Biosystems Inc., Foster City, CA, USA). Nucleotide numbering was based on the cDNA RefSeq sequence, with the A of the ATG start codon designated as +1. Following Vatsis nomenclature [44], the wild-type fast alleles (F) were classified as *NAT*2*4, *NAT*2*12, or *NAT*2*13, which encode normal *NAT*2 enzyme activity. Mutant alleles with reduced enzyme activity, designated as slow alleles (S), include *NAT*2*5, *NAT*2*6, *NAT*2*7, and *NAT*2*14. Study participants were categorized as homozygous fast (FF), heterozygous intermediate (FS), or homozygous slow (SS) acetylators based on their allele combinations.

### 4.4. Primary Outcomes

Primary outcomes were the percentage of patients reaching the recommended INH C_max_ and AUC_0–24h_ ranges described in the literature. The reference range for INH C_max_ was set at 3 to 6 mg/L [17]. For total INH exposure, we used the median AUC_0–24h_ range from studies included in a systematic review—11.6 to 26.3 h∙mg/L, excluding one study with outlier values—as a benchmark [18], following the approach used by Chabala et al. [25]. All clinical and laboratory adverse events were also collected.

### 4.5. Pharmacokinetic Parameters and Statistical Analysis

Pharmacokinetic parameters were estimated using non-compartmental analysis with WinNonLin (version 8.6, Pharsight Corp., Mountain View, CA, USA). The highest observed serum concentration was recorded as C_max_, with T_max_ representing the corresponding sampling time. Given the limited sampling points, the log-linear phase (log C versus t) was defined using two data points (3 and 6 h or 4 and 8 h). The absolute value of the slope (β/2.303, where β is the first-order elimination rate constant) was obtained through linear regression. The elimination half-life (*t*_1/2_) was calculated as 0.693/β. Concentrations beyond 6 or 8 h and up to 24 h were estimated based on decay of concentrations based on first-order pharmacokinetics described by the formula C_T2_ = C_T1_ × e^−β×(T2−T1)^. The area under the concentration-time curve (AUC_0–24h_ ) was determined using the linear-up/log-down trapezoidal rule from time zero to the last concentration at 24 h. Apparent clearance (Cl/F, where F represents bioavailability) was calculated as dose/AUC_0–24h_, while apparent volume of distribution (Vd/F) was derived as (Cl/F)/β.

Pharmacokinetic parameters were reported as medians and interquartile ranges (IQR). Categorical comparisons by gender, patient age at pharmacokinetic sampling (<2 years, Group A, or ≥2 years, Group B), *NAT*2 genotype, and concomitant use of other anti-TB medications were conducted using the Mann–Whitney *U* test or Kruskal–Wallis test, as appropriate. Associations between numerical variables were assessed via Spearman’s rank correlation coefficient. All statistical analyses were performed using SPSS version 24.0 (IBM Corp., Armonk, NY, USA), with statistical significance set at a *p*-value of <0.05. OpenAI’s ChatGPT (GPT-4-turbo) was used for language refinement to enhance clarity and coherence in preparing this manuscript.

## 5. Conclusions

In summary, in infants and children receiving a daily INH suspension at 10 mg/kg, no safety concerns were raised, and the recommended target adult C_max_ and AUC_0–24h_ levels were achieved in the majority of cases. AUC_0–24h_ values below or above the recommended range were associated with *NAT*2 acetylator status. Pharmacogenetic-driven dosing strategies may prove useful in the future to improve clinical outcomes and minimize toxicity, especially among the most vulnerable patients, such as those with severe disease, malnutrition, or HIV co-infection.

## Figures and Tables

**Figure 1 antibiotics-14-00074-f001:**
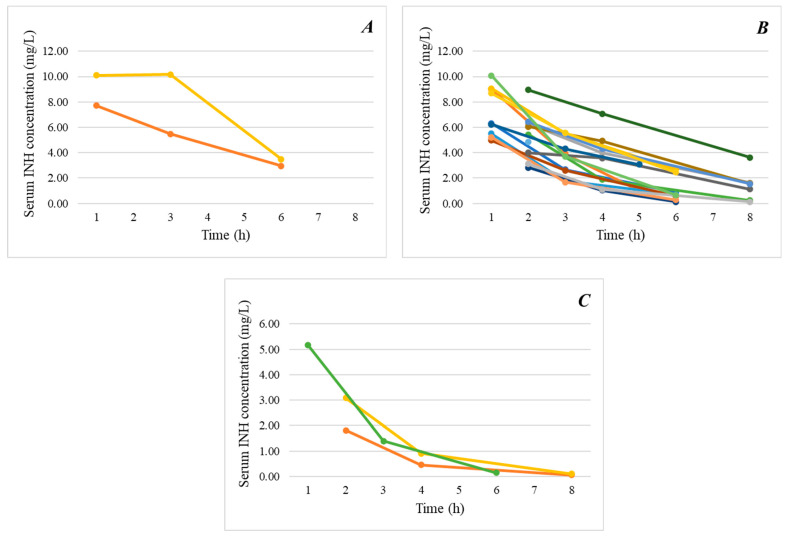
Isoniazid serum concentrations (in mg/L) according to *NAT*2 genotyping by acetylator type. (**A**) *NAT*2 homozygous slow genotype; (**B**) *NAT*2 heterozygous intermediate genotype; and (**C**) *NAT*2 homozygous fast genotype. Each line represents one patient.

**Table 1 antibiotics-14-00074-t001:** Clinical characteristics of the patients who completed the study protocol (*n* = 24); data are shown either as *n* (%) or as median (interquartile range), unless stated otherwise.

**At inclusion (*n* = 24)**	
Sex (female)	12 (50.0)
Ethnicity	
Latin American	12 (50.0)
Caucasian	5 (20.8)
Maghrebi	4 (16.7)
Asian	2 (8.3)
Black	1 (4.2)
Indication for INH treatment	
Primary chemoprophylaxis	12 (50.0)
TB infection	3 (12.5)
TB disease ^a^	9 (37.5)
ALT levels (IU/L)	19.5 (16.0–27.5)
Hemoglobin levels (g/dL)	12.4 (11.6–13.2)
Albumin levels (g/L)	44 (43.5–46)
**On the day of pharmacokinetic sampling (*n* = 24)**	
Age (years)	2.8 (1.8–4.2)
<2 years of age, Group A	7 (29.2)
≥2 years of age, Group B	17 (70.8)
Weight (kg)	13.3 (11.9–18.0)
Weight-for-length/height Z-score in patients < 5 years (mean, SD)	0.60 (0.86)
Body mass index-for-age Z-score in patients ≥ 5 years (mean, SD)	0.03 (0.45)
Malnutrition ^b^	0 (0)
Breastfeeding ^c^	2 (8.3)
INH dose (mg/kg)	10 (10.0–10.0)
Concomitant medications	
Other anti-TB drugs (rifampicin in all cases)	11 (45.8)
Steroids	2 (8.3)
Time on INH treatment (weeks)	12.4 (10.1–25.3)
ALT levels (IU/L)	21.0 (15.5–27.3)
ALT levels > 50 IU/L	3 (12.5)
Pre-dose fasting time (minutes)	681 (591–760)
Post-dose fasting time (minutes)	30 (30–35)

ALT, alanine aminotransferase; INH, isoniazid; SD, standard deviation; TB, tuberculosis. ^a^ Including pulmonary TB (*n* = 7) and TB meningitis and ankle arthritis (*n* = 1, one each). ^b^ Defined as a weight-for-length/height Z-score below −2 in children < 5 years of age, and as a body mass index-for-age Z-score below −2 in children > 5 years of age. ^c^ Neither of the two breastfeeding mothers was receiving INH.

**Table 2 antibiotics-14-00074-t002:** Summary of estimated pharmacokinetic parameters in 23 patients; one patient was excluded from this analysis because only the blood sample at the first time point was obtained. All data are shown as median (interquartile range).

	Median (IQR)	Patients Below/Within/Above the Recommended Range in Adults (In Brackets)
C_max_ (mg/L)	6.1 (4.5–8.2)	3/8/12 (3–6) [17]
T_max_ (h)	1.6 (1.2–2.0)	
AUC_0–24h_ (h∙mg/L)	23.0 (11.2–35.4)	6/8/9 (11.6–26.3) [18]
*t*_1/2_ (h)	1.7 (1.3–2.9)	
Cl/F (L/h)	5.7 (4.4–12.5)	
Vd/F (L)	21.7 (13.8–24.4)	

AUC_0–24h_, area under the concentration-time curve during the dosing interval; Cl/F, apparent clearance of the drug; C_max_, maximum drug concentration; IQR, interquartile range; *t*_1/2_, half-life; T_max_, time to C_max_; Vd/F, apparent volume of distribution.

**Table 3 antibiotics-14-00074-t003:** Effect of different clinical characteristics on isoniazid pharmacokinetic parameters in 23 patients; one patient was excluded from this analysis because only the blood sample at the first time point was obtained. All data are shown as median (interquartile range). All data analyzed using the Mann-Whitney U test, except for *NAT*2 genotype (Kruskal–Wallis test).

	*n*	C_max_ (mg/L)	*p*	T_max_ (h)	*p*	AUC_0–24h_ (h∙mg/L)	*p*	*t*_1/2_ (h)	*p*	Cl/F (L/h)	*p*	Vd/F (L)	*p*
Female	12	6.1 (4.9–9.0)	0.443	1.8 (1.3–2.0)	0.630	24.7 (12.4–35.3)	0.566	1.7 (1.3–3.0)	0.608	5.0 (4.0–9.7)	0.487	20.6 (12.6–24.3)	0.651
Male	11	5.3 (4.4–6.7)		1.6 (1.2–2.0)		15.7 (10.8–36.7)		1.7 (1.3–2.7)		7.2 (4.8–12.5)		21.8 (15.6–25.0)	
<2 years	7	6.3 (5.9–8.4)	0.234	1.2 (1.1–1.5)	0.075	23.0 (15.1–37.4)	0.624	1.8 (1.5–2.9)	0.413	4.6 (3.0–5.9)	0.585	13.8 (11.9–14.3)	0.003
≥2 years	16	5.2 (4.0–6.4)		2.0 (1.4–2.0)		20.0 (9.8–35.3)		1.5 (1.3–2.9)		8.2 (5.0–16.0)		22.9 (21.1–27.4)	
*NAT*2 genotype SS	2	8.9 (8.3–9.5)	0.030	2.1 (1.6–2.5)	0.592	50.0 (48.1–51.9)	0.011	2.8 (2.4–3.2)	0.026	2.4 (2.3–2.5)	0.006	9.9 (8.0–11.8)	0.024
*NAT*2 genotype FS	17	6.1 (5.0–8.1)		1.5 (1.1–2.0)		24.4 (15.2–35.3)		1.8 (1.3–3.0)		5.0 (4.6–9.9)		21.7 (13.8–23.2)	
*NAT*2 genotype FF	3	3.1 (2.4–4.1)		2.0 (1.7–2.0)		8.3 (6.5–9.3)		1.2 (1.1–1.2)		17.5 (16.9–27.5)		29.0 (26.7–48.1)	
INH monotherapy	12	5.5 (5.0–6.4)	0.865	2.0 (1.3–2.0)	0.531	16.4 (12.4–33.3)	0.786	1.6 (1.3–2.2)	0.651	6.8 (4.3–10.5)	0.928	18.5 (13.4–23.5)	0.608
INH + rifampicin	11	6.1 (3.6–8.2)		1.4 (1.3–2.0)		25.0 (10.2–37.3)		2.6 (1.3–3.0)		5.0 (4.4–14.4)		21.8 (16.2–26.6)	

AUC_0–24h_, area under the concentration-time curve during the dosing interval; Cl/F, clearance of the drug; C_max_, maximum drug concentration; FF, homozygous fast; FS, heterozygous intermediate; INH, isoniazid; *NAT*2, N-acetyltransferase 2; SS, homozygous slow; *t*_1/2_, half-life; T_max_, time to C_max_; Vd/F, apparent volume of distribution.

**Table 4 antibiotics-14-00074-t004:** Details of the adverse events in the 26 patients that received at least one dose of the study drug, according to age at enrollment.

	Total (*n* = 26)	Group A, <2 Years (*n* = 9)	Group B, ≥2 Years (*n* = 17)
Patients	8/26 (30.8%)	3/9 (33.3%)	5/17 (29.4%)
Any adverse event	11	5	6
Upper respiratory tract infection	4	3	1
Bronchospasm	1	1	0
Headache	1	0	1
Diarrhea	1	0	1
Pneumonia	1	0	1
Fever	1	0	1
Cough	1	0	1
Vomiting	1	1	0

## Data Availability

The original contributions presented in this study are included in the article. Further inquiries can be directed to the corresponding author.

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
