# Peer review of "Pharmacokinetic Analysis of an Isoniazid Suspension Among Spanish Children Under 6 Years of Age"

_antibiotics, 2025, doi:10.3390/antibiotics14010074_

Round 1
Reviewer 1 Report
Comments and Suggestions for Authors
This is a very interesting study given that there are few protocols for the treatment of TB in children. I suggest its publication once the authors address the observations and questions raised.
To the authors.
This is considered a well-founded project by the authors in favor of supporting TB treatment in children. There is a lot of information missing about the pharmacokinetic behavior and its application in the adjustment of INH doses in this age range.
I only make a few observations and questions to the authors:
Although the authors provide information on this subject, it is important to highlight how they determined the adherence to treatment of their patients? There are various tests that are applied for this purpose.
Were the adverse events recorded related to the NAT2 acetylator genotype? Did the researchers perform any statistical test on this subject?
What results could they have found in the case of carrying out this same study in a high TB ​​burden area in Spain? Is it not common for this health problem to occur in your country?
How did they determine the interindividual variability of Cmax and AUC?
Please justify the lack of measurement of AcetyINH through your chromatographic method. It is currently necessary to quantify this metabolite to study its relationship with the acetylator genotype and phenotype.
Did nursing staff measure the volume of the INH suspension prior to its administration?
Although information is provided in the document, how was the limited sampling scheme chosen?
Author Response
This is a very interesting study given that there are few protocols for the treatment of TB in children. I suggest its publication once the authors address the observations and questions raised.
This is considered a well-founded project by the authors in favor of supporting TB treatment in children. There is a lot of information missing about the pharmacokinetic behavior and its application in the adjustment of INH doses in this age range.
We thank the reviewer for the kind words and the suggestions made.
I only make a few observations and questions to the authors:
Although the authors provide information on this subject, it is important to highlight how they determined the adherence to treatment of their patients? There are various tests that are applied for this purpose.
In our outpatient clinic, adherence to antiTB treatment is assessed routinely within a nurse-led intervention that includes two monitoring steps (Eidus-Hamilton test and follow-up questionnaires), as explained in reference 10. For this particular study, parents were also asked to complete an ad hoc questionnaire the 3 days before the PK curve. This has been better explained in Methods (line 306).
Were the adverse events recorded related to the NAT2 acetylator genotype? Did the researchers perform any statistical test on this subject?
Thanks for raising this point. Adverse events or the development or hypertransaminasemia were not related to the NAT2 genotype. This information has been added to the Results (line 162)
What results could they have found in the case of carrying out this same study in a high TB ​​burden area in Spain? Is it not common for this health problem to occur in your country?
Thank you for the question. Tuberculosis is not highly prevalent in Spain, and there are no high TB burden areas within the country. This study was conducted in a low TB burden setting, which reflects the overall epidemiological context of Spain. Conducting the study in a high TB burden setting could potentially yield different results due to variations in factors such as nutritional status, prevalence of comorbidities (e.g., HIV), and genetic differences in drug metabolism. However, as high TB burden areas do not exist in Spain, this context was not applicable to our study population.
How did they determine the interindividual variability of Cmax and AUC?
Interindividual variability of Cmax and AUC was assessed using descriptive statistics (comparing individual values to mean/median values of the cohort), data stratification (acetylator genotype, age < or >2 years, and isoniazid monotherapy vs combined therapy), and appropriate statistical analysis.
Please justify the lack of measurement of AcetyINH through your chromatographic method. It is currently necessary to quantify this metabolite to study its relationship with the acetylator genotype and phenotype.
Thank you for highlighting this important point.
The metabolite acetyl-isoniazid is not active but serves as a valuable marker for phenotypically assessing acetylator status. This is typically done by evaluating the AcINH/INH ratio at a specific time point after dosing (doi: 10.2217/pgs.13.230). Alternatively, acetylator status can be determined phenotypically by calculating the half-life of the parent compound, isoniazid (doi: 10.1111/tmi.12003).
In our study, we did not assess acetylator phenotype because we performed NAT2 genotyping, which provides a more precise determination of acetylator status. This approach aligns with our methodology in a previous study (doi: 10.3390/antibiotics12020272).
Did nursing staff measure the volume of the INH suspension prior to its administration?
Yes, the study nurse was in charge of measuring the isoniazid dose to administer. This has been clarified in Methods (line 316).
Although information is provided in the document, how was the limited sampling scheme chosen?
Thank you for pointing this out. The limited sampling scheme was chosen based on a balance between obtaining sufficient data for reliable pharmacokinetic analysis and minimizing the burden on young participants. The scheme was informed by prior pharmacokinetic studies in pediatric populations, considering the need to capture key parameters such as Cmax and AUC while limiting the number of blood draws to ensure patient safety and comfort. We have clarified this in the manuscript to provide additional context (line 255).
Reviewer 2 Report
Comments and Suggestions for Authors
1. Should there be some comparisons to Bedaquiline? (published clinical work in this general area)?
2. I realize there are limitations in a study of this nature but is the data set small? Could data have been collected in countries with lower quality health care systems?
3. Table 4 is good, I think testing children with different illnesses is critical.
Author Response
- Should there be some comparisons to Bedaquiline? (published clinical work in this general area)?
Thank you for your comment. Bedaquiline is a drug used primarily for the treatment of multidrug-resistant tuberculosis and has a different mechanism of action and pharmacokinetic profile compared to isoniazid, which remains a first-line drug for drug-susceptible TB. Our study specifically focuses on characterizing the pharmacokinetics of a licensed INH suspension in young children, and given the space limitations, we did not address unrelated comparisons. However, we note that there are published studies on bedaquiline pharmacokinetics in children, particularly from high-burden TB regions, which might provide useful context for its use in pediatric multidrug-resistant tuberculosis treatment.
- I realize there are limitations in a study of this nature but is the data set small? Could data have been collected in countries with lower quality health care systems?
Thank you for your observations. Our study was conducted in Spain, a low TB burden country, where access to high-quality health care allows for close monitoring and rigorous pharmacokinetic analysis. While we acknowledge that the data set is relatively small, the study was designed as a phase IIa clinical trial with the primary aim of characterizing pharmacokinetics in a carefully selected population under standardized conditions. Conducting similar studies in high TB burden countries with diverse healthcare settings could complement our findings, particularly by addressing variations in malnutrition, comorbidities, and other factors that might influence isoniazid pharmacokinetics. This has been emphasized in the Introduction (line 91).
- Table 4 is good, I think testing children with different illnesses is critical.
Thanks for the interesting comments, which we did our best to address in the new version of the manuscript that now reads better and will be of further interest for the readers of the Journal.
Reviewer 3 Report
Comments and Suggestions for Authors
I appreciate the opportunity to revise this study that evaluates the pharmacokinetics of INH suspension in children under 6 years of age. Although the article does not provide many novelties about INH pharmacokinetics, it gathers knowledge about alternative formulations to treat patients with limitations in swallowing tablets, like children. The article is clear and well-written, but some important clarifications are remaining.
Table 4 and lines 159-160 are inconsistent with the information of other manuscript paragraphs and need checking. Line 101 mentions 27 children were enrolled, while Table 4 and Line 159 say 26. Line 159 mentions that the study recorded 12 mild adverse events, but Table 4 shows 11 instead. Line 159 and Table 4 mention: “8 out of 26 patients (29.6%)” which is 30.77% instead; however, 8 out of 27 is 29.63%
Tables 3 and 4 needs be renumbered.
Consider Replace the sentence: “None was considered to be causally related to the study drug” to “but all adverse events were considered not related to the treatment” in Lines 160-161.
The sentence: “The observed lower AUC0-24h in adults (11.6–26.3 h*mg/L) was not achieved in 6/23 (26.1%) patients” in Lines 137-138 is unclear; consider replace it by “the recommended AUC0-24h range in adults (11.6–26.3 h*mg/L) were not achieved in 6/23 children (26.1%).
Consider replace: “h*mg/L” by “h∙mg/L “ in all AUC units.
The relationship between the FF NAT2 acetylation genotype and the 6 children that have not achieved the adults recommended AUC0-24h range (11.6–26.3 h*mg/L) as mentioned in Lines 138-139 and 229-230 needs more details e.g., how was this hypothesis tested? Besides that, 3 children among this 6 are not FF NAT2 acetylation genotype. Would they be FS NAT2?
I suggest that the authors consider evaluating whether the body weight influences the pharmacokinetics of INH suspension in children. Because, besides the NAT2 genotype, body weight may be the most important anthropometric characteristic influencing the INH PK.
Author Response
I appreciate the opportunity to revise this study that evaluates the pharmacokinetics of INH suspension in children under 6 years of age. Although the article does not provide many novelties about INH pharmacokinetics, it gathers knowledge about alternative formulations to treat patients with limitations in swallowing tablets, like children. The article is clear and well-written, but some important clarifications are remaining.
Thanks for the kind comments and the suggestions, which have been addressed in the new version of the manuscript.
Table 4 and lines 159-160 are inconsistent with the information of other manuscript paragraphs and need checking. Line 101 mentions 27 children were enrolled, while Table 4 and Line 159 say 26. Line 159 mentions that the study recorded 12 mild adverse events, but Table 4 shows 11 instead. Line 159 and Table 4 mention: “8 out of 26 patients (29.6%)” which is 30.77% instead; however, 8 out of 27 is 29.63%
Apologies for these errors and thanks for spotting them.
Initially, 27 children were enrolled but 3 of them withdrew informed consent before PK day; 2 out of 3 received the study drug, however. AEs were recorded in 26 children who received the study drug.
Eleven (not 12) AEs were recorded, this has been corrected, and % have also been corrected. We hope this is now right and clear.
Tables 3 and 4 needs be renumbered.
Thanks for pointing this out. We are not sure what to do, since current Table 4 is cited earlier in the text than Table 3 but, because of formatting issues, Table 3 is inserted in the text after Table 4. We did not find specific instructions on how to address this in the Journal Instructions for Authors. What does the Editor suggest? No changes.
Consider Replace the sentence: “None was considered to be causally related to the study drug” to “but all adverse events were considered not related to the treatment” in Lines 160-161.
The sentence has been rephrased and shortened (line 160).
The sentence: “The observed lower AUC0-24h in adults (11.6–26.3 h*mg/L) was not achieved in 6/23 (26.1%) patients” in Lines 137-138 is unclear; consider replace it by “the recommended AUC0-24h range in adults (11.6–26.3 h*mg/L) were not achieved in 6/23 children (26.1%).
The sentence has been rephrased (line 137).
Consider replace: “h*mg/L” by “h∙mg/L “ in all AUC units.
Thanks again. The unit has been revised and corrected throughout the manuscript.
The relationship between the FF NAT2 acetylation genotype and the 6 children that have not achieved the adults recommended AUC0-24h range (11.6–26.3 h*mg/L) as mentioned in Lines 138-139 and 229-230 needs more details e.g., how was this hypothesis tested? Besides that, 3 children among this 6 are not FF NAT2 acetylation genotype. Would they be FS NAT2?
These results have been clarified in the text (line 138) and were available for 22 of the 24 patients, as the NAT2 genotype was unavailable for one patient and the AUC could not be estimated for another. The reviewer is correct that, apart from the three children with the FF NAT2 genotype, all other patients who did not reach the lower AUC range reported in adults were of the FS NAT2 genotype.
I suggest that the authors consider evaluating whether the body weight influences the pharmacokinetics of INH suspension in children. Because, besides the NAT2 genotype, body weight may be the most important anthropometric characteristic influencing the INH PK.
Thank you for this insightful suggestion. We agree that body weight can influence the pharmacokinetics of INH, particularly in the presence of malnutrition or extreme deviations from normal weight-for-age. However, all patients in our study had body weights within the expected range for their age, and no cases of malnutrition were observed. Given this homogeneity, we did not stratify the analysis by body weight, as we deemed it unlikely to introduce significant variability in the pharmacokinetic parameters. We have now added a clarification in the Limitations paragraph (line 256) to explain this decision.
Round 2
Reviewer 3 Report
Comments and Suggestions for Authors
no additional comments